# The Geometric World of Fishes: A Synthesis on Spatial Reorientation in Teleosts

**DOI:** 10.3390/ani12070881

**Published:** 2022-03-30

**Authors:** Greta Baratti, Davide Potrich, Sang Ah Lee, Anastasia Morandi-Raikova, Valeria Anna Sovrano

**Affiliations:** 1CIMeC, Center for Mind/Brain Sciences, University of Trento, 38068 Rovereto, Italy; davide.potrich@unitn.it (D.P.); a.morandiraikova@unitn.it (A.M.-R.); 2Department of Brain and Cognitive Sciences, Seoul National University, Seoul 08826, Korea; sangah@gmail.com; 3Department of Psychology and Cognitive Science, University of Trento, 38068 Rovereto, Italy

**Keywords:** navigation, spatial geometry, reorientation, teleosts, fishes

## Abstract

**Simple Summary:**

Animals inhabit species-specific ecological environments and acquire knowledge about the surrounding space to adaptively behave and move within it. Spatial cognition is important for achieving basic survival actions such as detecting the position of a food site or a mate, going back home or hiding from a predator. As such, animals possess multiple mechanisms for spatial mapping, including the use of individual reference points or positional relationships among them. One such mechanism allows disoriented animals to navigate according to the distinctive geometry of the environment: within a rectangular enclosure, they can simply reorient by using “metrics” (e.g., longer/shorter, closer/farther) and “sense” (e.g., left, right) attributes. Navigation based on the environmental geometry has been widely investigated across the animal kingdom, including fishes. In particular, research on teleost fish has contributed to the general understanding of geometric representations through both visual and extra-visual modalities, even vertebrates phylogenetically remote from mammals.

**Abstract:**

Fishes navigate through underwater environments with remarkable spatial precision and memory. Freshwater and seawater species make use of several orientation strategies for adaptative behavior that is on par with terrestrial organisms, and research on cognitive mapping and landmark use in fish have shown that relational and associative spatial learning guide goal-directed navigation not only in terrestrial but also in aquatic habitats. In the past thirty years, researchers explored spatial cognition in fishes in relation to the use of environmental geometry, perhaps because of the scientific value to compare them with land-dwelling animals. Geometric navigation involves the encoding of macrostructural characteristics of space, which are based on the Euclidean concepts of “points”, “surfaces”, and “boundaries”. The current review aims to inspect the extant literature on navigation by geometry in fishes, emphasizing both the recruitment of visual/extra-visual strategies and the nature of the behavioral task on orientation performance.

## 1. Foundations of Navigation by Geometry

One of the most significant skills for an animal to live a well-adapted life is to navigate to and from essential places in a goal-directed manner. The precise encoding of locations within an animal’s habitat is critical, as that is where animals carry out most behaviors essential to their survival, such as finding food, mating, and breeding. Such a capacity entails the simultaneous processing of multiple kinds of spatial cues and the implementation of complementary orientation mechanisms, such as the use of dead reckoning, celestial compass, magnetic field, landmarks, cognitive maps, and environmental geometry to guide navigation (see reviews: [1,2,3,4,5]). Some spatial navigation strategies involve detecting, processing, and memorizing specific sets of spatial cues, according to “egocentric” or self-based coordinates, while other involve “allocentric” mechanisms that rely on spatial relationships in world-based coordinates. 

Local landmarks as direct cues to aid in localization are grounded on sensory feedback, based on which simple stimulus-response associations (e.g., “food in the red-marked arm of the maze”) allow animals to execute goal-oriented movements (e.g., approach the red color) [6,7,8,9].

On the other hand, a cognitive map is a mental representation of a layout of multiple cues and the spatial relationships among them [10,11,12,13]. Allocentric strategies (e.g., place learning) and egocentric ones (e.g., guidance, cue learning), constitute two independent systems—the “local system” and the “taxon system” [14]—that can be selectively damaged, giving rise to place-cue learning dissociations (rats: [15,16,17]; birds: [18,19] reptiles: [20,21,22] fish: [23,24,25,26,27]).

Over the past thirty years, research on cognitive maps and landmarks as the basis for relational and associative learning have inadvertently led to environmental geometry as a focal issue in the field of spatial cognition and navigation. 

Imagine an experiment where you are standing in the center of a rectangular white room and in one corner there is a box containing a mystery prize. You are then blindfolded and slowly disoriented, during which time the box is quietly removed. Afterwards, you are asked to remove the blindfold and identify the corner where you saw the box before. If the experimental space was perfectly rectangular, rather than choosing one of the four corners at random or always choosing the correct corner, you will make systematic errors towards the 180° symmetric corner that have the same metric (short/long) and sense (left/right) attributes (see Figure 1) as the correct corner. This behavioral pattern will become increasingly consistent with repeated experience.

Cheng first [28,29] discovered geometry-biased behaviors in rats (*Rattus norvegicus*), observing that they searched for the treat either at the correct and at the rotationally symmetric corner of a rectangular arena, even in the presence of other cues that broke the room’s symmetry (see Figure 2). Cheng also designed a set of experiments to test the use of distinctive landmarks in conjunction with the geometric frame and compare the behavior of untrained versus trained rats in working memory (spontaneous choice) versus reference memory (learning behavior) tasks, respectively. In spontaneous choice tasks, animals usually perform a single session of test in which they are required to approach the location of a social object no longer present. By contrast, in learning behavior tasks, animals are rewarded after approaching the correct position (e.g., the two geometrically equivalent corners) until they meet a certain criterion of correctness. One of the main distinctions relates to experience, which is absent in working memory tasks, thus, not decisive to reorient. 

Cheng [28] introduced two tests aimed at detecting how geometry and features could be combined to specify the rewarded locations: the “diagonal transposition”, in which the landmarks marking the rewarded diagonal were switched; the “affine transformation”, where each landmark was moved 90° right to put the features in conflict with geometry. Results can be summed up as follows: (1) both in working and reference memory tasks, rats made systematic rotational errors, signifying the successful use of geometry; (2) in the reference memory task only, rats incorporated the nongeometric cues and chose the correct corner; (3) after removing the landmarks from the two geometrically correct corners, rats could not distinguish the symmetric corners by taking advantage of features that are distant from them; (4) after the affine transformation, rats persisted in choosing the correct geometry as observed during the pre-test phase. Cheng advanced the hypothesis that two independent cognitive systems are engaged in the spatial reorientation behavior of rats: a “geometric module” for encoding metric and sense properties of surfaces and “featural subsystems” for computing nongeometric information. Despite the dissociable representation, these processes were purported to work in synergy for navigation and reorientation. 

From Cheng’s investigations surfaced a strict conception of the geometric module, consistent with the modularity theory proposed by Fodor [30]. Following subsequent studies in various species, the idea of an encapsulated navigation system—at least at the level of behavior—was revised, and several alternative models were proposed to explain the wide range of results coming from behavioral to neurophysiological, to neuroimaging studies [31,32,33].

The “modularity revised” view [34,35,36], for instance, revisits the original ideas by Cheng [28] and Gallistel [37], combined with a view of multiple navigation systems put forward by Doeller and Burgess [38]. This approach considers a wide range of studies on animal behavior, human development, functional neuroimaging, and neurophysiology (see review: [39]). It claims that three-dimensional layouts and features are cognitively dissociable and separately implemented in the vertebrates’ brain. In this respect, several studies have shown that whereas boundary-based navigation is associated with the hippocampal entorhinal cortex and subiculum, landmark-based navigation is instead associated with the striatal basal ganglia [40,41,42,43]. Moreover, this view is supported by recent rodent data on the role of environmental geometry in the alignment of hippocampal place maps [44], as well as direct neural recordings in both rodents and humans showing activity of the entorhinal cortex [45,46] and subiculum [42,47] near physical boundaries (e.g., walls of the enclosures).

With the wide range of tasks and results concerning environmental boundary-based geometric navigation in vertebrates (see reviews: [48,49]) and invertebrates [50,51,52,53]), it is difficult to build a cohesive view that addresses all facets of the problem. However, the importance of geometry as a core spatial concept shared widely across the phylogenetic tree (see review: [54]) comes from behavioral investigations in fish, which represent a large branch of vertebrates phylogenetically remote from mammals.

## 2. Navigation by Visual Geometry in Fishes

Fish populates a diverse range of ecological niches. Just like terrestrial animals, fish face complex spatial problems, planning and executing adaptive movements to remembered goals [55]. Not surprisingly, fishes have been shown to possess remarkable abilities in spatial mapping and navigation (see for instance: [56,57,58,59,60]; see reviews: [61,62]). Although teleosts have long been classified as one of the most primitive vertebrates, and often considered to have very limited cognitive capacities (see for instance: [63]), nowadays it is becoming increasingly evident that these aquatic organisms have learning and memory processes comparable to those displayed by land tetrapods (see review: [64,65,66]). Beyond such core abilities, an even finer-grained distinction can be made with regards to species-specific adaptations. In the early 2000′s, growing amount of converging evidence collected in various animal species on geometric reorientation behavior piqued the interest of researchers across many fields of study [67,68,69].

One interesting study on navigation by geometry in fish, by Sovrano and colleagues [70], adapted the reference memory introduced by Cheng [28] to train a fish species, *Xenotoca eiseni*, to reorient within a rectangular white tank. Basically, the operant conditioning procedure (a “rewarded exit task”) required fish to locate the correct corner at which there was an open exit leading to an outer tank with food and conspecifics. Fish learned to reorient not only according to the arena’s shape, but also in conjunction with a conspicuous landmark (i.e., a blue panel placed on one of the four walls). This result demonstrates the integrated use of geometric and nongeometric cues in fish.

In a follow-up study, Sovrano and colleagues [71] further explored the use of local landmarks in relation to spatial geometry by adding a pattern-specific panel at each corner of a rectangular white arena. Male and female *X. eiseni* were trained by using the same procedure as above, then performing four probe tests (i.e., “removal of all panels”, “removal of the two geometrically-correct panels”, “diagonal transposition”, and “affine transformation”), to clarify the role of geometry and features, particularly when they were put in conflict. Results can be summed up as follows: after removing all panels, fish showed a consistent preference for the correct-geometry diagonal; after removing the two panels on the correct-geometry diagonal, only females persisted in choosing the geometry while males were completely unable to reorient, choosing at random the four corners of the arena; after the diagonal transposition, fish focused on the local panel cue as learned during training, in a new position but geometrically correct; after the affine transformation, the behavioral choice was split: moving the local cues 90° clockwise created a conflict between geometric and featural information, such that fish followed both the local landmark (in the new position) and the two geometrically-correct corners (i.e., as defined by the previous panel’s original position). 

Around the same time, Vargas and colleagues [72] were investigating the use of geometric and featural spatial information in another fish species, the goldfish *Carassius auratus*. Likewise, fish were subjected to an extensive training within a rectangular arena, both with and without an additional nongeometric cue (i.e., an angular panel covering two adjacent walls). Results were close to those by Sovrano et al. [70,71], although after the affine transformation (here called “dissociation test”), goldfish significantly chose in accord with the feature instead of geometry, again showing the independent encoding of the geometry and feature, especially apparent when the feature directly marked the goal. 

Furthermore, the research team of Vargas [73] also found a place-cue learning dissociation during geometric navigation after damaging the lateral pallium of goldfish (the assumed homologue of the mammalian hippocampus). Fish with lateral pallium lesions, in contrast with those with medial pallium lesions and sham controls, were unable to use the arena’s spatial geometry and, instead, just relied on the landmark. For the first time, the involvement of the fish telencephalon in reorientation was demonstrated, drawing a parallel with analogous evidence in other vertebrates [74,75]. Consistent with this, a study by Rajan and colleagues [76] showed that active spatial learning within a rectangular arena induced, in the telencephalon of goldfish, the expression of immediate-early gene (IEG) early growth response 1 (egr-1), which is a regulatory transcription factor involved in neural plasticity and memory formation in mammals (see review: [77]).

One perplexing issue in the complementary effects of geometry and landmark was related to the size of the experimental space (see review: [48]). Sovrano and colleagues [78,79] ran several training experiments within rectangular arenas of different sizes, providing the usual blue panel as a conspicuous landmark. They found that while *X. eiseni* were able to combine shape and feature in both arenas, when the cues were placed in conflict as in the case of the affine test, fish mainly relied on shape in the smaller environment (where rotational errors were dominant) and on feature in the larger one (where errors near the proximal landmark were dominant). A possible explanation concerns the “visual scanning” hypothesis [80], which states that in small spaces animals can extract geometry in its totality (seeing simultaneously both the longer and the shorter surface), whereas in large spaces animals can only get a partial view of the metric information (not seeing the longer surface in full). On the other hand, an alternative interpretation grounded on observing a strong effect of previous experience with feature in large arenas on subsequent reorientation behavior (see for instance: [81]) pointed to the existence of a weighted combination strategy based on adaptive learning [82,83].

All the experiments on fish described so far employed a reference memory procedure: animals were trained over time until a fixed learning criterion (usually, greater than or equal to 70% of correctness), by rewarding them in the case of success (as purported by the operant conditioning). Given that the original demonstration of geometric primacy was found in unrewarded working memory tasks, in both rats and human children [28,84], more recent studies engaged a working-memory-like procedure focused on spontaneous choices made by fish in “socially cued” tasks. This paradigm uses a simple visual observation of a conspecific as a social attractor to set the target location: during the encoding phase, the subject is trapped in a transparent container at the center of the arena and the conspecific, also in a transparent container, is placed at one of the corners of the arena (e.g., rectangular or square); during the disorientation procedure, the social stimulus is removed, and the experimental subject’s subsequent approaches to the four corners are observed. The findings were astonishingly in line with the original studies in mammals. 

For instance, Lee and colleagues [85] first developed and tested the above procedure with *D. rerio* and *X. eiseni* within a rectangular and a square arena with one blue panel as a landmark. They found that both species, in the absence of rewarded training, chose the two geometrically correct corners. However, there were several additional effects associated with sex and distance in *X. eiseni* (i.e., the proximity of the landmark as regards the target): (1) only males used both geometry and feature, in contrast to findings by Sovrano et al. [71]; (2) the feature was useful only when close to the goal and not when it was far from it (in that case, fish reoriented in accord with the arena’s shape); (3) the forementioned landmark combination was only observed in *X. eiseni* (reorientation behavior of *D. rerio* primarily geometry-dependent).

Although *X. eiseni* has proven to be an excellent animal model to investigate navigation behavior, the zebrafish *D. rerio*, following Streisinger’s pioneering investigations [86], has risen as a popular model in neuroscience (see for instance: [87]) and biomedical research (see review: [88]). 

Lee and colleagues [89] assessed in zebrafish the representation of specific geometric components (i.e., distance, corners, and length) that make up an environment’s shape, and their relation to directional sense (i.e., left/right) in a spatial layout. In this study, the social cued reorientation task was performed inside a transparent rectangular or square arena, partially covered with white opaque panels on the walls or corners to provide boundary-related visual information. The use of distance (from the center to the walls) was evaluated by adding four white panels of equal length to the center of each wall of a transparent rectangular arena; the use of corners (the point where two walls converge) was evaluated by adding four L-shaped white covers to the rectangle’s corners; the use of length (of the walls, as opposed to the distance between them) was evaluated by adding two long white panels and two short white panels on opposing walls of a transparent square arena. Untrained fish spontaneously reoriented according to distance (i.e., closer/farther surfaces on the left/right) but searched randomly when provided with only the corner array or the walls of different length arranged in a square array. Similar results were previously found in 2-year-old children in reorientation tasks within segmented rhombic and rectangular arrays [90]. Baratti and colleagues [91] recently trained zebrafish under the experimental conditions defined by Lee and colleagues [89]. Different groups of fish underwent the rewarded exit task, and each of them was presented with one of the three boundary-related arrays as described above. Besides the distance condition, it was observed that fish learned to use both corners and length over time, thus suggesting that they could represent all the geometrically informative components of a rectangular arena and that the behavioral task demand (i.e., the rewarded training) can recruit spatial learning mechanisms that affected the ability of zebrafish to extract the geometric cues provided by the experimental environment.

Lee and colleagues [92] also carried out several experiments in zebrafish aimed to examine the use of landmarks as beacons during spontaneous geometric navigation. In the absence of a visible geometrically informative context (i.e., within a rectangular transparent arena), zebrafish failed to merge geometry and feature under several conditions: in the presence of a two-dimensional visual form (i.e., a rectangular blue panel placed on the ground surface); with one large three-dimensional landmark (i.e., a blue cylinder in a transparent arena), which only served as a local landmark; when the feature was distal from the target, even if the cue was a light source on one end of the arena. Interestingly, however, when the landmarks were embedded within a visible geometrically informative context (i.e., within a rectangular opaque arena), zebrafish chose the two corners along the correct-geometry diagonal despite the presence of a 3-D blue cylinder that broke the symmetry of the arena.

In order to clarify how the environmental geometry could provide a spatial framework for landmark-use and how proximity could interact with such a process, Sovrano and colleagues [93] investigated in *X. eiseni* the use of features (i.e., different panels at corners) in unrewarded social cued tasks, while varying the number and proximity of features with respect to the target, in both geometrically informative and noninformative spatial layouts (i.e., rectangle versus square). Improvements in landmark-use were investigated through a rewarded reference memory task in a square arena. Under spontaneous, unrewarded navigation tasks, visual landmarks acquired perceptive salience but without being a spatial cue to place-finding when distal from the goal. However, fish gradually overcame these limits over time, particularly when the visual landmarks were close to the goal (acting as beacons). Similar results were previously obtained in mice [94], with significant differences in landmark-use (relative to geometry) in spontaneous versus rewarded tasks. In the former condition, the landmark was used only to distinguish whether the goal was close to it, regardless of left-right information; but with reinforcement, in the latter condition, animals improved their use of the landmark to identify the target corner.

One detail to note is that the study by Sovrano et al. [93] was conducted with *X. eiseni*, while investigations by Lee et al. [92] were carried out with *D. rerio*. As reported above, the study by Lee et al. [85] found differences in landmark-use depending on the species. The issue of species differences in landmark-based reorientation should be directly faced through targeted experiments using both rewarded and spontaneous behavioral protocols. Evidence in this regard could help to resolve the debate on the interaction between geometry and landmark in navigation, at least in teleosts. 

As a preliminary step, Baratti and colleagues [95] started evaluating the use of pure geometry by subjecting a group of zebrafish to the rewarded exit task within a rectangular white arena. Consistently with training studies in *X. eiseni* [70] and *C. auratus* [72], *D. rerio* learned to encode the boundary for reorientation, with improved performance over time. Through this initial study, the effectiveness of the rewarded exit task as a training paradigm to assess relational learning in zebrafish was validated. Future efforts should perhaps be geared not only towards investigations on matching environmental geometry and local features (e.g., a conspicuous blue panel, pattern-specific panels), but also on landmark-use in the absence of geometric layouts (e.g., within a circular or square arena). Parallel to this, the encoding of absolute and relative distances in the presence of landmark arrays could provide further insight to the representation guiding spatial navigation behavior in zebrafish [96]).

In general, all of the studies on boundary- and landmark-based reorientation behaviors seem to support, also in fishes, the existence of two underlying independent mechanisms: a “spatial system” for the processing of geometric information (in terms of invariable properties of the global environment, such as distances and directional sense relations), and a “landmark system” for the processing of landmark-based information (in terms of landmarks or objects that may be more variable or more useful as a local cue). These two systems would be differently recruited to guide behavior in spontaneous or rewarded navigation, especially with respect to what is adaptive as a cue to the target in the particular task at hand. An important and intriguing issue is the possible neurobiology of disoriented navigation behavior. As mentioned above, three-dimensional layouts and features are dissociable systems, which are distinctly implemented in the brain of vertebrates (e.g., hippocampus versus striatum). Hence, reported homologies between the hippocampus of mammals and the lateral pallium of teleosts [45,97], together with a common organization of the basal ganglia [98], lead us to believe that boundaries and features could be functionally dissociable also in the behavior of disoriented fish. Amniotes and teleosts may also share the same molecular mechanisms underlying hippocampal function. For instance, a study by Gómez and colleagues [99] revealed that map-like spatial representations in goldfish could be compromised by blocking hippocampal N-methyl-D-aspartate (NMDA) receptors. In goldfish, the recruitment of the dorsolateral telencephalon has been demonstrated multiple times in the use of geometry [73,100]. Following their initial study [73], Vargas and colleagues [100] went further into the investigation of specific telencephalic ablations on geometric navigation in the absence of other featural cues. Differently from previous results, they noticed that lateral pallium lesions did not entirely impair the ability of goldfish to reorient by geometry when the goal position was unambiguous (two escape corners instead of one). Although many studies investigating the neural correlates of spatial cognition in teleosts converge upon the finding that allocentric hippocampus-dependent mechanisms are largely independent from cue-learning [23,101,102,103,104,105,106,107], it is possible that some redundant processing of the same set of cues occurs across multiple mechanisms during navigation (e.g., a corner could be represented both as a local visual cue, as well as a part of a global environmental geometry). More recent evidence further suggests a differential involvement of hippocampal pallium subregions of the goldfish’ brain over the course of spatial memory formation [108,109,110]. 

Lastly, two other issues have been investigated in geometric navigation of trained fish: the impact of brain lateralization [111] and the role of rearing conditions, both in *X. eiseni* [112]. Results from the former study showed that lateralized fish achieved better performance than nonlateralized ones at merging geometry and feature (i.e., a salient blue panel) and at using pattern-specific landmarks in the absence of metric attributes (i.e., within a square arena). On the other hand, the latter study showed that the encoding of an enclosed space’s shape is “inborn” and independent from early experience inside geometric (rectangular home tank) or nongeometric (circular home tank) environments. Similar findings had been previously obtained by Brown and colleagues [113] with convict fish, in which controlled rearing conditions with or without featural cues affected the relative influence of landmarks, but not the ability to use geometry alone. 

All the reported evidence described so far seems to strengthen the boundary-based view also in fishes, similarly to data collected in other vertebrates (see review: [39]). Nevertheless, most of these experiments were run within rectangular characterized by three-dimensional visible surfaces, usually consisting of four white or black opaque walls. While the role of nonvisual geometry started being evaluated in recent years ([114], see below for details), reorientation skills in environments of other shapes have not been addressed fully in fish. Another issue is related to potential differences in behavior due to appetitive versus aversive tasks both in spontaneous and rewarded reorientation problems. For instance, a couple of studies in rats have shown that aversive conditions (a water maze task; see also: [15,16]) improved the capacity of animals to orient, both in working [115] and reference [116] memory tasks. Specifically, in regard to geometric reorientation, Golob and Taube [115] observed that animals observed in a rectangular water maze with one landmark wall made less rotational errors than those observed in a dry maze. If aversive contexts lead animals to preferentially use an “escape strategy” based on nongeometric cues, as put forward by the authors, then the variability in landmark-use may reflect the preferential recruitment of hippocampal or striatal mechanisms, in appetitive or aversive tasks, respectively [117]. Lastly, it could be interesting to design new experiments targeted to handle other geometric variables by employing new experimental protocols (for instance, like in Twyman et al. [118]), to test the generalization of the existing evidence on the geometry-plus-landmark interplay, while also taking task demands into consideration.

## 3. Navigation by Nonvisual Geometry in Fishes

Experimental data on navigation by visual geometry in fishes are quite clear about the role of well-defined boundaries in navigation, both in spontaneous and rewarded tasks. However, the sensory nature of processes recruited when animals regain heading and position, or search for salient hidden objects still remains an unresolved issue. Studies with insects [52,53,119,120], rats [121,122], and birds [123,124], have suggested the involvement of view-based strategies during reorientation behaviors. Some claim that navigation by geometry is guided by visual sequential “snapshots” of the environment and that animals move to reduce pixel-by-pixel mismatches between retinal and stored images [122]. Nevertheless, such a global matching theory may not explain evidence from vertebrate navigation, as in studies by Lee and Spelke [125] with 44-month-old toddlers and by Lee and colleagues [126] with newly-hatched chicks, showing that successful reorientation occurred within even visually subtle three-dimensional layouts that confined the exploratory space, but not within visually salient two-dimensional forms or arrays of objects. 

Visual encoding of geometric information has been well-documented in humans (toddlers: [90,127,128]) and nonhumans [89,129] and is also supported by neuroimaging and neurostimulation studies that generally revealed the recruitment of high-level visual processing regions like the parahippocampal place area (PPA), the occipital place area (OPA), and the retrosplenial cortex (RSC) in the presence of macrostructural extended surfaces such as occluding barriers [38,130,131,132,133,134,135,136,137,138]. Although the involvement of visual mechanisms seems to be strongly interlaced with boundary-based navigation, this interplay may be subjected to a dissociation of vision from tactile physical access, allowing animals to perceive nonvisible barriers as obstacles, despite the lack of visual stimulation. 

Some developmental and comparative data on the use of non-view-based strategies during reorientation may speak to this fact. A study by Gianni and colleagues [139] found that children aged 5–7 were able to use transparent walls to navigate within a rectangular room, but that younger children aged 2–5 were not able to do so. This study showed that representation of boundary structures before the age of five seem to heavily rely on visually-driven mechanisms. However, perhaps with the accumulation of multisensory experience of the surrounding world, the effect of nonvisual barriers on navigation emerges later in development.

At the same time, a study by Sovrano and colleagues [114] found conceptually similar evidence in fishes. Two hypogean species of fish (*Astyanax mexicanus* and *Phreatichthys andruzzii*, blind “cavefish”) underwent the traditional rewarded exit task developed by Sovrano et al. [70]. Despite their lack of vision, cavefish learned to navigate by geometry, also by combining it with a tactile landmark (a panel with embossed stripes) for successful place-finding. From a conjectural point of view, these results signify that in organisms where the visual system is somehow impaired, boundary-based navigation towards a salient goal can be driven by extra-visual stimulation, such as haptic exploration [140]. Then, such an intriguing idea could be put into practice by exploring the reorientation capacities of blind people, with the potential to observe the role of neural plasticity on disoriented navigation. 

The recruitment of extra-visual systems extends to the behavior of eyed species, even those that primarily use sight to represent space, as the engagement of other sensory systems than vision would aid in the precise encoding of location through the addition of environmental cues. 

Sovrano and colleagues [141] recently addressed the use of nonvisual geometry in three fishes (*D. rerio*, *X. eiseni*, and *C. auratus*), by employing both the unrewarded social cued task and the rewarded exit task described previously, within a rectangular transparent arena to directly assess the encoding of nonvisual geometry in the absence of any other visual features. While the first two experiments were run within a similar arena with respect to Lee and colleagues [85,89,92], the third experiment introduced exit corridors for the training task, modified, for the transparent conditions (see Figure 3). 

This study showed a noteworthy difference in reorientation by nonvisible boundaries which, like landmarks, depended on the task procedures and, therefore, on the memory system employed. Fishes could not spontaneously reorient (both without and with a short period of familiarization) but learned to use the transparent boundaries in the reference memory task. The extended training may have allowed fishes to freely swim within the arena and acquire extra-visual experience of the spatial contingencies in relation to the goal-position. Furthermore, these results support the engagement of other sensory modalities beyond the vision in spatial learning and memory. One of these modalities could be the “lateral line”: the function of this mechano-sensory system is detecting hydrodynamic stimuli, such as weak water movements and pressure gradients, allowing fish to navigate underwater [142,143,144,145,146,147,148,149,150,151,152,153]. A brief mention should be made to the electrosensory system of weakly electric fish, which is another rich and unique modality as an alternative to vision. Electric fish can interact with the environment and communicate with companions by means of “electric images”, which are a product of changes in the electric field due to fish-to-object proximity interactions [154,155]. Spatial learning abilities were observed in weakly electric fish [156,157,158], as well as hippocampal-like circuitry in the pallium [159]. These species could provide a new animal model to explore extra-visual geometric navigation, since neural algorithms and computations underlying the transduction of sensory contrasts are argued to be shared and preserved among senses [160]. 

Main results on navigation by visual and nonvisual environmental geometry in fishes are summarized in Table 1.

## 4. Concluding Remarks

Although the entirety of the extant data cannot be explained within a single all-encompassing model, the converging evidence that boundary-based core representations of spatial geometry and adaptively learned landmark-based navigation provide a strong argument for shared systems of spatial cognition across vertebrates, including both fish and mammals. Nevertheless, fish are a powerful animal model for studying the neurobiological and adaptive origins of different modalities and additional perceptual systems that have been sculpted by the aquatic habitat or visual conditions in deep underwater environments.

## Figures and Tables

**Figure 1 animals-12-00881-f001:**
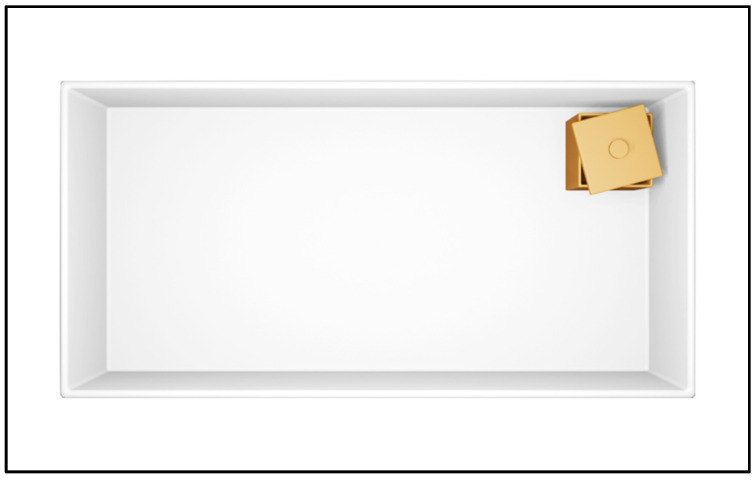
Within a rectangular white room, the corner with the box has a long surface on the left and a short one on the right (metric and sense attributes). The same is true for the location that is 180° rotationally symmetric corner. These two corners are geometrically identical, and the other two corners are characterized by opposite geometry (a long surface on the right and a short one on the left).

**Figure 2 animals-12-00881-f002:**
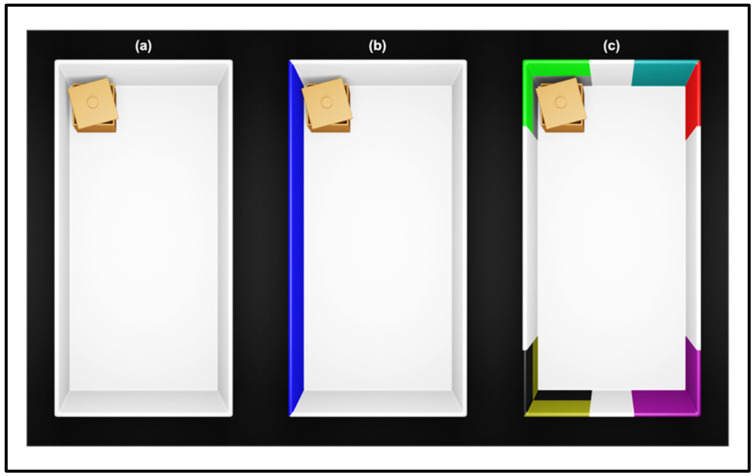
The geometric symmetry in (**a**) can be resolved by providing distinctive nongeometric cues. In (**b**), one of the two long surfaces has been painted in blue, and the correct corner can be identified by integrating the geometry with the landmark (i.e., the target corner has a long blue surface to its left). In (**c**), each corner has a distinctive pattern/color, and the local landmark itself is enough to identify the target (i.e., the corner with the box is lime-gray patterned).

**Figure 3 animals-12-00881-f003:**
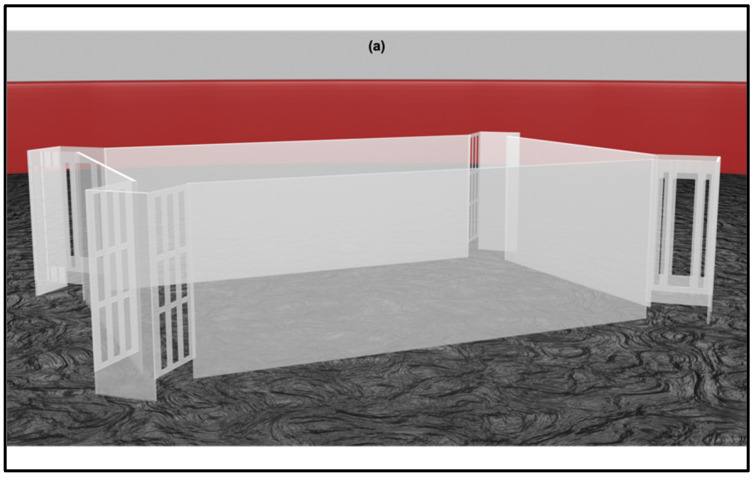
The figure shows the rectangular transparent arena adapted for the nonvisual rewarded exit task (**a**). Four corridors were placed at the corners – two with a large central slit that fish could exit through (correct-geometry diagonal corners) and the other two with smaller slits that fish could not swim through (**b**).

**Table 1 animals-12-00881-t001:** Summary of major findings on geometric navigation by fishes, within visual and nonvisual spatial layouts. Working and reference memory tasks are specified to distinguish across behavioral protocols, visual and nonvisual to distinguish across experimental modalities.

Studies	Major Results
Sovrano et al., 2002 [70]	In a reference memory task in visual modalities, *X. eiseni* learn to use both the rectangular geometry and the blue wall to reorient.
Sovrano et al., 2003 [71]	In a reference memory task in visual modalities, *X. eiseni* show a preference for geometry after the all-panels removal; for the trained local landmark after diagonal transposition; for geometry and the local landmark after the affine transformation, even in conflict. Some sex-specific differences found after the correct-panels removal (only females use geometry).
Vargas et al., 2004 [72]	In a reference memory task in visual modalities, *C. auratus* learn to use both the rectangular geometry and a corner landmark to reorient but show a preference for the landmark after affine transformation.
Sovrano et al., 2005 [78]	In a reference memory task in visual modalities, *X. eiseni* mainly reorient by geometry if trained in a small arena and tested in a big one and use the blue wall if trained in a big arena and tested in a small one.
Sovrano et al., 2005 [111]	In a reference memory task in visual modalities, lateralized *X. eiseni* is better at combining geometry and the blue wall, and at using local landmarks in the absence of metric attributes.
Vargas et al., 2006 [73]	In a reference memory task in visual modalities, *C. auratus* with lateral pallium lesions do not use geometry to reorient and just rely on the landmark.
Sovrano et al., 2007 [79]	In a reference memory task in visual modalities, *X. eiseni* mainly reorient with geometry in a small arena and with the blue wall in a big arena, after affine transformation of big landmarks (blue walls)
Brown et al., 2007 [113]	In a reference memory task in visual condition, controlled rearing conditions with or without featural cues affect the influence of landmarks, but not the ability to use geometry alone, in convict fish (*A. nigrofasciatus*).
Vargas et al., 2011 [100]	In a reference memory task in visual modalities, *C. auratus* with lateral pallium lesions are not totally impaired at using geometry to reorient when the target can be unambiguously located.
Lee et al., 2012 [85]	In a working memory task in visual modalities, *X. eiseni* and *D. rerio* use the rectangular geometry in the absence of training. Some species- and sex-specific differences have been found at simultaneously using geometry and the blue wall (females find harder the disengagement from geometry).
Lee et al., 2013 [89]	In a working memory task in visual modalities, *D. rerio* reorient according to boundary distance and sense but not by corners or boundary length.
Lee et al., 2015 [92]	In a working memory task in nonvisual modalities, *D. rerio* fail to merge several kinds of features with the geometry of a transparent rectangular arena. Some effects of proximity found in relation to the target position.
Sovrano & Chiandetti, 2017 [112]	In a reference memory task in visual modalities, *X. eiseni* reared within circular tanks reorient just as well as fish reared within rectangular tanks. The encoding of environmental geometries is “inborn” and independent from early experience
Sovrano et al., 2018 [114]	In a reference memory task in nonvisual modalities, hypogean *A. mexicanus* and *P. andruzzii* learn to use both the rectangular geometry and a tactile landmark with embossed stripes to reorient
Sovrano et al., 2020 [141]	In working and reference memory tasks in nonvisual modalities, *X. eiseni*, *D. rerio*, and *C. auratus* learn to use nonvisual geometry only over time under rewarded training (but not in the absence of training), probably relying on extra-visual sensory modalities. The different outcome of the geometric reorientation is strongly based on the type of experimental procedure.
Sovrano et al., 2020 [93]	In working and reference memory tasks in visual modalities, *X. eiseni* use features only to determine if the target is close regardless of metric attributes but overcome this limit over time under rewarded training.
Baratti et al., 2020 [95]	In a reference memory task in visual modalities, *D. rerio* learn to use the rectangular geometry to reorient, also showing improvements over time.
Baratti et al., 2021 [91]	In a reference memory task in visual modalities, *D. rerio* learn to use both corners and boundary length, in addition to distance combined with sense, to reorient.

## Data Availability

Not applicable.

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
