# Peer review of "The Geometric World of Fishes: A Synthesis on Spatial Reorientation in Teleosts"

_animals, 2022, doi:10.3390/ani12070881_

Round 1
Reviewer 1 Report
Dear authors/editor(s)
It is obvious that the authors put a considerable amount of work into improving the manuscript. I took the liberty to make some direct comments in the PDF, all of them are highlighted in yellow. Those sections marked in yellow that lack an additional comment simply indicate sections where some language editing is required.
Section I introduces some core concepts of navigation based on geometry and is much improved. However, it still suffers from imprecisions that should be addressed in a final round of edits. Please refrain from figurative speech and be as precise in your terminology as possible to avoid confusing the readers not aware of the original literature!
Section II provides a lot of experimental studies supporting the dual role of feature vs. geometry in fish. As above I find the flow of arguments, the sequence of studies (i.e, presenting results along species) difficult to follow, as it leads to a – in my impression ! – highly convoluted text. The section on a possible distinction between hippocampus like area-mediated spatial learning and the potential contribution of the more conserved BD/pallial regions was interesting though.
Section III addresses non-visual geometric learning, leaning strongly towards non-fish results. The informative table should be improved by spelling out the conditions rather than relying on abbreviations.
Section iV further addresses non-visual based learning, specifically behaviours attributed to the mechanosensory LL. As before, the reader does not need to learn more about this sensory system than that it detects local flow fields, the authors obviously are no experts on the subject (excuse my frankness) of the anatomy/physiology here and would avoid several imprecisions by not diving into the subject too deep. In the short section providing info on the electric fish, citations should be those of studies actually demonstrating spatial learning, including recent pallial electrophysiological studies rather than very general studies on behavioural capabilities of these fishes.

Author Response
It is obvious that the authors put a considerable amount of work into improving the manuscript. I took the liberty to make some direct comments in the PDF, all of them are highlighted in yellow. Those sections marked in yellow that lack an additional comment simply indicate sections where some language editing is required.
We thank the reviewer for the suggestions. We now improved all sections that were highlighted in yellow.
Section I introduces some core concepts of navigation based on geometry and is much improved. However, it still suffers from imprecisions that should be addressed in a final round of edits. Please refrain from figurative speech and be as precise in your terminology as possible to avoid confusing the readers not aware of the original literature!
The imprecisions were fixed and a more consistent terminology was used throughout the text as suggested by the Reviewer.
Section II provides a lot of experimental studies supporting the dual role of feature vs. geometry in fish. As above I find the flow of arguments, the sequence of studies (i.e, presenting results along species) difficult to follow, as it leads to a – in my impression ! – highly convoluted text. The section on a possible distinction between hippocampus like area-mediated spatial learning and the potential contribution of the more conserved BD/pallial regions was interesting though.
We thank the Reviewer for this suggestion. However, we decided to report fish studies mainly by following a chronological order, which also delineate two subsections as regards the behavioral tasks (rewarded vs. spontaneous).
Section III addresses non-visual geometric learning, leaning strongly towards non-fish results. The informative table should be improved by spelling out the conditions rather than relying on abbreviations.
We are thankful to the Reviewer for pointing this out, we know that nonvisual evidence discussed in section III also refer to other species than fish. The aim was to introduce an issue (i.e., reorientation by nonvisual geometry) not so explored in fish but with a potential for more targeted studies, since the opportunity to assess the use of extra-visual sensory modalities.
We improved the informative table as suggested by the Reviewer.
Section iV further addresses non-visual based learning, specifically behaviours attributed to the mechanosensory LL. As before, the reader does not need to learn more about this sensory system than that it detects local flow fields, the authors obviously are no experts on the subject (excuse my frankness) of the anatomy/physiology here and would avoid several imprecisions by not diving into the subject too deep. In the short section providing info on the electric fish, citations should be those of studies actually demonstrating spatial learning, including recent pallial electrophysiological studies rather than very general studies on behavioural capabilities of these fishes.
We considered to delete the whole section IV to avoid confusing the readers with information not so related to the main topic of this review (i.e., present effective findings and not speculate about further studies). A brief mention to the lateral line and electric fish was provided at the end of session III.
Reviewer 2 Report
The authors have made several improvements to the manuscript. All concerned has been adequately addressed. Rearranging the introduction has helped the writing’s flow. And some relevant topics have been included in the discussion.
Author Response
The authors have made several improvements to the manuscript. All concerned has been adequately addressed. Rearranging the introduction has helped the writing’s flow. And some relevant topics have been included in the discussion.
We are very thankful to the Reviewer for this positive comment.
This manuscript is a resubmission of an earlier submission. The following is a list of the peer review reports and author responses from that submission.
Round 1
Reviewer 1 Report
The review entitled “The geometric world of fishes: A systematic review on navigation in teleosts” by Baratti et al. has the potential to be a comprehensive and review of navigation by geometry in fishes. In a nutshell, I think in the present form this is not yet the case as the text lacks focus and the wordy sections one to three are not well connected with the intended major focus of the actual “fish-part” (sections 4&5). As a side note, the title should be changed to better reflect the intended focus of the manuscript, which clearly is not “navigation in fish” in general.
Chapter 6 focusses on the lateral line system as an interesting modality to investigate the contribution of non-visual sensory systems to navigation. This, including the interesting studies reviewed by the authors, is certainly correct. However, the authors fail to mention research on the electrosensory lateral line system, a modality equally unique as it requires stitching together egocentric coordinates with allothetic sensory input of a classical near-range sensory system. Also, the description of the mechanosensory lateral line system is not totally correct (e.g.: the ALLN, that is the anterior lateral line nerve conveys sensory input of both canal and superficial neuromasts from the head (see 587), nor are the sensory cells responsible for transduction called air cells).
I consider the data reviewed starting with chapter 4 highly relevant and would urge the authors to focus on these important aspects while taking care to revise the writing to be fully comprehensible to non-experts while defining the different concepts clearly. That being said, I’ll only point out some aspects I noted while going through the manuscript. There were many more passages that would require clarifications/revision of language before the manuscript can be accepted. However, I think it is first necessary to re-arrange and re-focus the review to the truly novel aspects, before more detailed feedback to the authors is sensible.
Notes:
The fact that fish vary tremendously in the niches they inhabit does not allow to connect this abundance with the need of spatial knowledge: „Animal species inhabit a wide range of ecological environments, making it necessary for them to acquire knowledge about the surrounding space in order to adaptively behave and move within it.”
In the abstract the authors refer to cognitive maps as the roof of spatial learning. I don’t think the debate on the actual presence of a Tolman-sense cm is finished and it needs to be shown that the cm is at the base of fish navigation for sure, before this generalization is acceptable.: Cognitive maps and landmarks are at the root of relational and associative spatial learning, guiding
Line 58: “at the starting position 2,” should be to the…
Lines 59 ff: Here the authors try to disambiguate between ego- and allocentric navigation. Accordingly, they define egocentric as: “based on inertial coordinates”. I fail to understand what is being proposed here and I sincerely doubt that other readers will gain the crucial distinction between ego- and allocentric navigation strategies from this paragraph. Rewriting is required. I think this is true for the whole section where the authors try to distinguish various terms used in navigation research.
Line 128, correct “two corners has the 50% of correctness”
Correct sentence starting at line 134.
The whole paragraph aiming to introduce the studies by Cheng is highly convoluted and again I doubt that readers will get the essentials.
Author Response
The review entitled “The geometric world of fishes: A systematic review on navigation in teleosts” by Baratti et al. has the potential to be a comprehensive and review of navigation by geometry in fishes. In a nutshell, I think in the present form this is not yet the case as the text lacks focus and the wordy sections one to three are not well connected with the intended major focus of the actual “fish-part” (sections 4&5). As a side note, the title should be changed to better reflect the intended focus of the manuscript, which clearly is not “navigation in fish” in general.
Our intent for 1-3 sections was to provide a general framework to the main topic (geometric navigation by fish), especially for naïve readers who know a little about spatial reorientation. We aimed to give a kind of didactic approach to the manuscript, by reporting also well-known concepts for people focusing on navigation at large as major research interest. In order to minimize the amount of redundant information, we reduced the first two introductory sections, by unifying them within one single section. We further think that the Manuscripts’ sections are well balanced in terms of form and content. With respect to the title, our new proposal is: “The geometric world of fishes: A systematic review on spatial reorientation in teleosts”.
Chapter 6 focusses on the lateral line system as an interesting modality to investigate the contribution of non-visual sensory systems to navigation. This, including the interesting studies reviewed by the authors, is certainly correct. However, the authors fail to mention research on the electrosensory lateral line system, a modality equally unique as it requires stitching together egocentric coordinates with allothetic sensory input of a classical near-range sensory system. Also, the description of the mechanosensory lateral line system is not totally correct (e.g.: the ALLN, that is the anterior lateral line nerve conveys sensory input of both canal and superficial neuromasts from the head (see 587), nor are the sensory cells responsible for transduction called air cells).
According to the Reviewer, we mentioned the electro-sensory lateral line (lines 681-695). Moreover, we corrected the inaccuracies about the mechano-sensory lateral line as well.
I consider the data reviewed starting with chapter 4 highly relevant and would urge the authors to focus on these important aspects while taking care to revise the writing to be fully comprehensible to non-experts while defining the different concepts clearly. That being said, I’ll only point out some aspects I noted while going through the manuscript. There were many more passages that would require clarifications/revision of language before the manuscript can be accepted. However, I think it is first necessary to re-arrange and re-focus the review to the truly novel aspects, before more detailed feedback to the authors is sensible.
We thank the Reviewer for this suggestion, but we argue that the introductory sections we proposed comprise all the concepts needed to non-experts to grasp the topic. We simplified several paragraphs for a better understanding, trying to clarify essential aspects.
Notes:
The fact that fish vary tremendously in the niches they inhabit does not allow to connect this abundance with the need of spatial knowledge: „Animal species inhabit a wide range of ecological environments, making it necessary for them to acquire knowledge about the surrounding space in order to adaptively behave and move within it.”
It could be possible that in the absence of abundance and variability of habitats, as regards spatial cues, animals use a more limited set of strategies to “interpret” those cues. Anyway, we deleted the cause-effect link, by modifying both sentences at lines 9-10 and 275-278.
In the abstract the authors refer to cognitive maps as the roof of spatial learning. I don’t think the debate on the actual presence of a Tolman-sense cm is finished and it needs to be shown that the cm is at the base of fish navigation for sure, before this generalization is acceptable.: Cognitive maps and landmarks are at the root of relational and associative spatial learning, guiding
We replaced “are at the root of” with “seem to be relevant mechanisms for”.
Line 58: “at the starting position 2,” should be to the…
We corrected it.
Lines 59 ff: Here the authors try to disambiguate between ego- and allocentric navigation. Accordingly, they define egocentric as: “based on inertial coordinates”. I fail to understand what is being proposed here and I sincerely doubt that other readers will gain the crucial distinction between ego- and allocentric navigation strategies from this paragraph. Rewriting is required. I think this is true for the whole section where the authors try to distinguish various terms used in navigation research.
We rewrote and simplified the section.
Line 128, correct “two corners has the 50% of correctness”
We corrected it.
Correct sentence starting at line 134.
We corrected it.
The whole paragraph aiming to introduce the studies by Cheng is highly convoluted and again I doubt that readers will get the essentials.
We simplify the paragraph, according to the Reviewer’s suggestion.
Reviewer 2 Report
The review is well written and very informative. It basically discusses all major points related to the use of spatial geometric information by fish. Some suggestions that I think could improve the text:
-The ms devotes more than 5 pages to introduce spatial cognition and geometric cues. That extension is more appropriate for a book chapter that for a review.
- From my point of view, the information about the neural basis of spatial navigation in fish using geometric cues is incomplete. There are very few papers on the topic but the authors have included only the oldest one. In general, the more recent literature on the encoding of geometric information by goldfish is missing.
-There is no mention to the effect of the appetitive Vs aversive reinforcement. The topic was first deeply discussed on the paper: Golob EJ, Taube JS (2002) Differences between appetitive and aversive reinforcement on reorientation in a spatial working memory task. Beh Brain Res 136:309–316
-Given the didactic approach of the ms, I suggest summarizing the main findings on the topic at the end of the text. A table, figure or text could be enough.
Minor comments
Line 272-3. The sentence “Although teleosts have for long been classified as the most primitive vertebrates, and therefore considered to possess scant cognitive capacities” needs to be referenced.
Line 283-4. “The first study considering navigation by geometry in fish is attributable to Sovrano and colleagues”. This study was published in 2002 and two years before there was already a PhD thesis on this topic and a conference communication “C. Broglio, Y. Gómez, J.C. López, F. Rodríguez, C. Salas, J.P. Vargas. Encoding of geometric and featural properties of a spatial environment in teleostean fish (Carassius auratus). Int. J. Psychol., 35 (2000), 195-195.” Moreover, in 2001 there were also a couple of conference communications: “Sovrano V.A., Bisazza A., Vallortigara G. Spatial reorientation using geometric and featural properties of an environment by fish (Xenotoca eiseni). Advances in Ethology, 36 Supplements to Ethology, p.265 (Abstract Book, XXVII International Ethological Conference, 22-29 August 2001, Tübingen, Germany). Sovrano V.A., Bisazza A., Vallortigara G. Fish use of geometric and non-geometric properties of an environment for spatial reorientation. Behavioural Pharmacology, vol. 11, suppl.1, p. S97 (Abstract Book, The First Joint Meeting of the European Brain and Behaviour Society and The European Behavioural Pharmacology Society, 8-12 September 2001, Marseille, France).
Line 323. “[83,84,4]”. There is a typo on the cites or the numbers are nor organized.
Author Response
The review is well written and very informative. It basically discusses all major points related to the use of spatial geometric information by fish. Some suggestions that I think could improve the text:
The ms devotes more than 5 pages to introduce spatial cognition and geometric cues. That extension is more appropriate for a book chapter that for a review.
We reduced the first two introductory sections, by unifying them within one single section.
From my point of view, the information about the neural basis of spatial navigation in fish using geometric cues is incomplete. There are very few papers on the topic but the authors have included only the oldest one. In general, the more recent literature on the encoding of geometric information by goldfish is missing.
We added more recent papers about the neural basis of geometric navigation by goldfish (lines 475-487).
There is no mention to the effect of the appetitive Vs aversive reinforcement. The topic was first deeply discussed on the paper: Golob EJ, Taube JS (2002) Differences between appetitive and aversive reinforcement on reorientation in a spatial working memory task. Beh Brain Res 136:309–316
We discussed such an effect, as suggested by the Reviewer, at lines 510-521.
Given the didactic approach of the ms, I suggest summarizing the main findings on the topic at the end of the text. A table, figure or text could be enough.
We summarized the main findings on geometric navigation by fishes within a table (Table 1).
Minor comments
Line 272-3. The sentence “Although teleosts have for long been classified as the most primitive vertebrates, and therefore considered to possess scant cognitive capacities” needs to be referenced.
We referenced the sentence.
Line 283-4. “The first study considering navigation by geometry in fish is attributable to Sovrano and colleagues”. This study was published in 2002 and two years before there was already a PhD thesis on this topic and a conference communication “C. Broglio, Y. Gómez, J.C. López, F. Rodríguez, C. Salas, J.P. Vargas. Encoding of geometric and featural properties of a spatial environment in teleostean fish (Carassius auratus). Int. J. Psychol., 35 (2000), 195-195.” Moreover, in 2001 there were also a couple of conference communications: “Sovrano V.A., Bisazza A., Vallortigara G. Spatial reorientation using geometric and featural properties of an environment by fish (Xenotoca eiseni). Advances in Ethology, 36 Supplements to Ethology, p.265 (Abstract Book, XXVII International Ethological Conference, 22-29 August 2001, Tübingen, Germany). Sovrano V.A., Bisazza A., Vallortigara G. Fish use of geometric and non-geometric properties of an environment for spatial reorientation. Behavioural Pharmacology, vol. 11, suppl.1, p. S97 (Abstract Book, The First Joint Meeting of the European Brain and Behaviour Society and The European Behavioural Pharmacology Society, 8-12 September 2001, Marseille, France).
We added the references suggested by the Reviewer before discussing the study by Sovrano and colleagues (2002).
Line 323. “[83,84,4]”. There is a typo on the cites or the numbers are nor organized.
We reorganized the cites.